# Application of an ice sheet model to evaluate PMIP3 LGM climatologies over the North American ice sheets

Jay R. Alder<sup>1</sup>, Steven W. Hostetler<sup>1</sup>

<sup>1</sup>US Geological Survey, 104 CEOAS Admin Building, Corvallis, 97331, USA

Correspondence to: Jay R. Alder (jalder@usgs.gov)

**Abstract.** We apply the Community Ice Sheet Model (CISM2) to the Palaeoclimate Modelling Intercomparison 3 (PMIP3) Last Glacial Maximum (LGM) simulations to determine if the general circulation models (GCMs) simulated surface temperature and precipitation climatologies would support the large North American ice sheets. We force CISM2 with eight PMIP3 GCMs, and an additional model,

- GENMOM. The ice sheet simulations indicate seven GCMs produce LGM temperature and precipitation climatologies that support positive mass balances of the Laurentide and Cordilleran ice sheets (LIS, CIS) in areas corresponding to those prescribed in the GCMs, and two GCMs simulate July temperatures that are too warm to support the ice sheets. Four of the nine GCMs support the development of ice sheets in Beringia in the CISM2, in conflict with the driving GCM and reconstructions that indicate the area was
- ice-free. We test the sensitivity of our results over a range of snow and ice positive degree-day factors, and we evaluate the role of albedo, and shortwave and longwave radiation in the simulations. Areas with perineal snow in the GCM simulations are found to correspond well to the CISM2 simulation of ice presence.

# **1** Introduction

- The Palaeoclimate Modelling Intercomparison 3 (PMIP3) Last Glacial Maximum (LGM, 21,000 years ago) experiment, which is part of the Coupled Model Intercomparison Project Phase 5 (CMIP5, Taylor et al., 2012), provides an opportunity to evaluate the ability of general circulation models (GCMs) to simulate climate states different from the present. (Braconnot et al., 2012; Hargreaves et al., 2013; Harrison et al., 2014). The extent, height, and topography of the North American ice sheets (NAIS) are
- key boundary conditions for LGM model simulations that have been explored in numerous model

experiments (Braconnot et al., 2007; COHMAP Members, 1988; Harrison et al., 2016; Schmittner et al., 2011). The three phases of PMIP have included different ice sheet reconstructions that vary greatly in volume and surface topography (Abe-Ouchi et al., 2015; Peltier, 1994; 2004). The area and southern extent of the North American ice sheets (NAIS) is well constrained (Dyke, 2004; Dyke and Prest, 1987;

- 5 Ehlers, 2011), yet there is little direct evidence of the distribution of the ice mass. Geophysical models are used to estimate the ice sheet surface topography constrained by observed relative sea-level changes and post-glacial rebound rates (Abe-Ouchi et al., 2015; Marshall et al., 2002; Peltier et al., 2015). While the LGM climate is reasonably well constrained by data over land and oceans, climate model simulations are the only source of information for precipitation, temperature, and radiative balance over ice sheets
- 10 which control surface mass balance and hence ice mass and topography. The use of ice sheet configurations as boundary conditions and the resulting climate simulations in the PMIP3 project provide the opportunity to employ a stand-alone, dynamic ice sheet model to evaluate the climatologies over NAIS and the degree to which those climatologies support the area of the NAIS.
- In a previous evaluation of the PMIP models, Pollard (2000) used output from 10 PMIP1 models and found that four of the models produced positive mass balances over the Laurentide ice sheet (LIS), where the remaining six had unrealistically negative surface mass budgets. In a recent study, Ullman et al. (2015) applied a surface energy balance model to a LGM GCM simulation and found the LIS to have a positive mass balance in the interior of the ice sheet and strongly negative mass balance along the margins, resulting in a net positive budget. Here we evaluate the ice-sheet climatologies of eight PMIP3 models and one additional model by using the simulated temperature and precipitation fields to force a 3-
- D thermomechanical ice sheet model to simulate the absence or presence of simulated ice relative to the fixed ice sheets prescribed in the GCMs. Our goal is to use the ice sheet model to determine if the simulated temperature and precipitation climatologies support the NAIS.
- The paper is organized as follows: Section 2 describes the ice sheet model, initialization and 25 PMIP3 driving climatology; Section 3 reviews results of the ice sheet model simulations; Section 4 analyzes GCM scale snow, albedo and radiation fields to put the ice sheet simulations into the context of the larger GCM simulation; Section 5 concludes with a summary of our results and implications for future CMIP assessments.

# 2 Methods

#### 2.1 Ice sheet model and configuration

We apply the 3-D thermomechanical Community Ice Sheet Model 2.0.5 (CISM, Price et al., 2015;
Rutt et al., 2009). We drive CISM2 with nine PMIP3 GCMs through a one-way, offline, coupling. CISM2
5 is configured to use a shallow-ice approximation and annual positive degree-day (PDD) mass balance scheme (Reeh, 1991) such that surface melt is proportional to the sum of positive degree-days over a year. The model has been successfully applied to simulate the Greenland Ice Sheet (Lipscomb et al., 2013; Lunt et al., 2008; Stone et al., 2010) and the North American deglaciation (21 – 7ka) (Gregoire et al., 2012; 2015). In this application, we use monthly temperature and precipitation from the PMIP3 GCMs in the

- 10 annual PDD scheme. Daily air temperatures in the PDD scheme are assumed to follow a sinusoidal cycle based on mean-annual and July temperatures (Price et al., 2015). In the PDD scheme, the calculation of melt is sensitive to July temperature, as it defines the number of days above freezing (Reeh, 1991). All precipitation is assumed to fall as snow, and up to 60% of the snow melt can refreeze.
- Our model configuration largely follows that of Gregoire et al. (2012) who used the same 40-km 15 North American domain on a Lambert azimuthal equal area projection with a soft bed basal sliding parameterization for sediment (Table 1). The domain encompasses the LIS, Cordilleran Ice Sheet (CIS) and Greenland Ice Sheet. Gregoire et al. (2012) used PDD factors of 3 mm d<sup>-1</sup> °C<sup>-1</sup> for snow and 8 mm d<sup>-1</sup> °C<sup>-1</sup> for ice (Marshall et al., 2002); however, these values are often derived from modern-day Greenland observations and are not well constrained for the LIS (Pollard, 2000), and Hebeler et al. (2008), for example, found that the simulated ice extent of the Fennoscandian Ice Sheet to be sensitive to the choice of PDD factors. To address uncertainties in PDD factors, we explore the sensitivity of the NAIS simulations to a range of PDD factors in 12 combinations of [3, 4, 5] mm d<sup>-1</sup> °C<sup>-1</sup> for snow and [8, 12, 16, 20] mm d<sup>-1</sup> °C<sup>-1</sup> for ice (range suggested by Stone et al., 2010), resulting in a total of 108 simulations for all the GCMs. A constant lapse rate of 5 °C km<sup>-1</sup> (Abe-Ouchi et al., 2007; Gregoire et al., 2012) is used
- 25 to adjust temperature between the GCM topography and CISM2 surface height.

The large volume of the NAIS complicates initializing CISM2 to LGM conditions. One approach is to spin-up the model with modern topography and build up ice through the last glacial–interglacial cycle using a climate-index technique (Charbit et al., 2007; Gregoire et al., 2012; Marshall and Clarke,

1999; Marshall et al., 2002; Zweck and Huybrechts, 2005). A second approach is to use modern topography with a eustatic sea level adjustment, which is subject to large lapse corrections. We opted for a third alternative: to initialize CISM2 with a surface height similar to the GCM LGM topography, which minimizes lapse adjustments. The ICE-6G\_C reconstruction (Argus et al., 2014; Peltier et al., 2015)
provides both ice thickness and orography at the LGM, which allowed us to initialize the NAIS in the CISM2 with reasonable ice volume, relative sea level changes and bedrock depression. We note that the blended CMIP5/PMIP3 ice sheet (Abe-Ouchi et al., 2015) has the same areal extent as ICE-6G\_C over the NAIS, but different surface topography. The North American change in ice volume (LGM minus present), expressed in eustatic sea level equivalent (SLE), is 80.5 m for ICE-6G\_C, which is greater than
the composite CMIP5/PMIP3 (78.6 m) and ICE-6Gv.2 (76.8 m), but within the range of other estimates (60.5 - 82.5 m, Abe-Ouchi et al., 2015).

We integrated CISM2 for 5000 years to ensure that the ice sheets had sufficient time to respond to the GCM forcing. A simulation length of 5000 years is roughly equivalent to the duration of the LGM (7500 years, 26.5 to 19 ka) proposed by Clark et al. (2009) during which global ice sheets were in near-

- 15 equilibrium. Although the simulated ice volumes were not in true equilibrium after 5000 years, the areal extents were largely stable. If the GCM simulated temperatures are too warm in areas with prescribed ice cover, the ice sheet model will ablate until the initialization ice is removed, resulting in a data-model mismatch. If the GCM simulated temperatures are too cold in areas prescribed to be land in the GCM, the ice sheet model will incorrectly accumulate ice, also a data-model mismatch. Thus, evaluating the absence
- 20 or presence of ice cannot constrain lower temperatures over the NAIS but can constrain the upper bound of temperature on ice grid cells and lower bound of temperature on land grid cells.

# 2.2 PMIP3 models and forcing climatology

We evaluate LGM simulations from eight PMIP3 models from the CMIP5 archive and GENMOM (Alder et al., 2011) (Table 2). While not part of PMIP3, we included GENMOM in our evaluation because
the boundary conditions used in the LGM simulation are similar to those of PMIP3 (Alder and Hostetler, 2015) and earlier versions of the GENESIS atmospheric model were included in previous PMIP evaluations (Joussaume et al., 1999; Pinot et al., 1999; Pollard, 2000).

We calculated long-term monthly climatological means from the available time series for each model over its respective grid. We use raw model output directly, without bias correcting the model output by an observed modern climatology (i.e. 'delta' method, Pollard, 2000). We drove CISM2 with the monthly values of near-surface air temperature (*tas*) and precipitation (*pr*). Our approach is 'one-way' in that simulated changes of the ice sheet do not feedback to the driving fields through attendant changes in albedo, the energy balance, or circulation. Surface temperatures vary through a lapse correction that adjusts for differences between the fixed GCM topography (*orog*) and the dynamic ice sheet surface height simulated by CISM2. Where available, we also obtained additional GCM fields, such as land mask (*sftlf*), ice mask (*sftgif*), snow area fraction (*snc*), snow depth (*snd*), surface shortwave, and longwave 10 fields (*rsds, rsus, rlds, rlus*) to broaden the analysis.

The expression of the ice sheet topography varies based on the resolution of the GCM: higher resolution models have a faithful implementation of the 1° x 1° CMIP5/PMIP3 ice sheet (Fig. 1). Two GISS-E2-R simulations have been submitted to the PMIP3 archive: one that uses ICE-5G and a second one that blends ICE-5G with a different reconstruction of the LIS surface height (Licciardi et al., 1998).

- 15 To drive CISM2, we chose the blended Licciardi et al. (1998) and ICE-5G realization because that simulation is similar to the CMIP5/PMIP3 surface topography. Differences in the height of the two LIS configurations have been shown to drive changes in planetary wave structure and atmospheric circulation (Ullman et al., 2014) which result in different temperature and precipitation climatologies over the LIS. Unlike the CMIP5/PMIP3 ice sheet, the GISS-E2-R configuration has a clear divide of lower elevation
- 20 between the LIS and the CIS. The GENMOM simulation also uses Licciardi et al. (1998) for the LIS topography, but blended with ICE-4G elsewhere (Table 2). Initializing GENMOM and GISS-E2-R with ICE-6G\_C rather than their respective ice sheet configurations does not alter the results.

The CNRM-CM5 and FGOALS-g2 model files have surface topographies that are inconsistent with their ice masks in that their masks indicate no ice over Hudson Bay. Their simulated climate fields appear to be reasonable, so the mask artifact may be a byproduct of the Coupled Model Output Rewriter (CMOR, http://www2-pcmdi.llnl.gov/cmor) postprocessor setting the ice mask to zero in grid cells that are ocean in the piControl experiment. The ice mask over the Fennoscandian Ice Sheet in both models supports this assessment.

### **3 Results**

#### **3.1 LGM climatology**

For all models the mean annual temperature (MAT) is below freezing in nearly all ice-covered grid cells, and many models simulate MATs below -30 °C over the LIS (Fig. 2, top). CNRM-CM5
displays the weakest cooling and IPSL-CM5A-LR the strongest. CNRM-CM5 also has the weakest global land-based cooling of the PMIP3 models (Harrison et al., 2015). With the exception of CNRM-CM, most of the models simulate July temperatures over the LIS well below freezing (Fig. 2, bottom). MRI-CGCM3 displays above-freezing temperatures along the southern margin of the LIS, a feature that CCSM4 also exhibits to a lesser degree. Nearly half the models (CCSM4, FGOALS-g2, GISS-E2-R, MIROC-ESM)
simulate areas of below freezing July temperatures over Beringia. In the PDD melt calculation, July

temperatures below freezing result in no annual melt.

All the models simulate similar mean-annual precipitation over NAIS (Fig. 3). CCSM4 and CNRM-CM5 display the wettest interiors and GISS-E2-R simulates substantial precipitation along the divide between the LIS and the CIS. Winters are dry in the interior and wet on the margins, particularly

15 along margins adjacent to the oceans. Precipitation penetrates further into the interior of the ice during summer, where precipitation totals are generally between 3 – 10 times greater than winter (not shown). Most of the models maintain summer temperatures below freezing, making summer the strongest accumulation season with 40 – 60% of the annual total precipitation occurring between June and August.

#### **3.2 Summary of PDD factors**

- A key issue underlying the simulations is establishing the sensitivity of steady-state ice sheets to the PDD factors. We chose to specify a set of values for snow and ice PDD factors in CISM2 that we used for all models. Most model simulations display relatively low sensitivity of the ice sheet extent to the PDD factors, with all 12 simulations for each GCM largely displaying the same ice sheet extent (Fig. 4). Ice mask differences between the suite of PDD factor simulations are limited to the margins. CCSM4,
- 25 GENMOM, and GISS-E2-R forced CISM2 simulations display southern boundaries that are well constrained by the GCM across all combinations of snow and ice factors. The CCSM4 and GISS-E2-R simulations display substantial ice over Beringia that is not specified as a boundary condition in the

GCMs, and the GENMOM and GISS-E2-R simulations display ice extents greater than prescribed in the GCMs along much of their northern margins. The southern margins in the IPSL-CM5A-LR, MIROC-ESM, and MPI-ESM-P simulations extend further south than is prescribed in the respective GCMs. Many of the PDD combinations in the FGOALS-g2 simulations produce ice that extends almost to the bottom

5 of the model domain. The prescribed position of the southern margin varies somewhat among the GCMs due to their varying resolutions. The four of the seven models that simulate a reasonable LIS also simulate greater ice-covered area over Beringia.

When forced by CNRM-CM5 or MRI-CGCM3, the CISM2 failed to simulate the NAIS comparable to the GCM boundary conditions, for any combination of PDD factors. All the CISM2 simulations were initialized with ICE-6G\_C volume and bedrock depression, but the initial ice volume is ablated in regions with negative mass balance in these two GCMs. Both models simulate July temperatures that are too warm to support the LGM ice. The MRI-CGCM3 simulations display a feedback wherein the negative mass balance along the southern margin reduces the elevation of the ice, which causes the surface temperature to warm through lapse rate correction as the difference between the GCM surface topography and the increasingly lower CISM2 surface height is amplified, which in turn drives

the MRI-CGCM3 southern margin further north.

Both ICE-6G\_C and the blended CMIP5/PMIP3 ice sheet include mountain glaciers and ice caps over the US Rocky Mountains and the Sierra Nevada. A number of higher resolution GCMs prescribe this ice as boundary conditions which provide an additional target to assess if the simulated temperature

- 20 and precipitation climatologies of the GCMs support permanent ice at higher elevations south of the NAIS margin. Both IPSL-CM5A-LR and MPI-ESM-P CISM2 simulations produce ice in the Sierra Nevada where it is specified in the GCMs. The GENMOM and GISS-E2-R CISM2 simulations display ice in the US Rocky Mountains even though the glaciers are not prescribe in the GCMs. Areas in US Rocky Mountains that should have isolated mountain glaciers are connected to the LIS/CIS in the FGOALS-g2,
- 25 IPSL-CM5A-LR, and MIROC-ESM simulations with most combinations of PDD factors, indicating the GCM temperatures are too cold or precipitation is too high, or both, at these lower latitudes.

5

#### 3.3 CISM2 simulated ice sheets

Seven out of nine GCMs simulate temperature and precipitation that produce a reasonable extent of the LIS. In many cases, the simulated southern margin is in close agreement with the prescribed ice sheets in the GCMs. The ice margins simulated by CISM2 are in general steeper and the domes are lower than ICE-6G\_C. The CISM2 develops maximum ice sheet thicknesses > 2800 m, which are ~800 m lower than the ICE-6G\_C that is > 3600 m in some regions (Fig. 5). After the first 100 years of CSIM2 simulation, the thickest areas initialized from ICE-6G\_C begin to thin and flatten, indicating that either the CISM2 physics or our choice of parameters do not support the domes present in ICE-6G\_C. The CISM2 includes a parameterization that scales ice advection. We conducted sensitivity tests with CISM2 to investigate whether the steep domes could be reduced by reducing the flow factor parameterization 10 from 5 to 1 which effectively reduces the viscosity of the ice but we found that the margins and dome

shapes were not sensitive to this parameter.

The GCMs that simulated below freezing July surface air temperatures over Beringia developed large ice volumes in this region after 5000 yrs. In a number of CISM2 simulations, the ice thickness 15 exceeds 2000 m which causes a corresponding bedrock depression. In the GENMOM CSIM2 simulations, a small area of positive mass balance develops over Beringia along the Arctic Ocean and the ice advects into adjacent grid cells with zero or slightly negative mass balance. The advected ice in these adjacent cells cools air temperatures through the lapse correction, which produces a net positive mass balance that is ultimately balanced by ablation thereby stabilizing the Beringian ice cover.

- 20 South-to-north cross section transects along the eastern portion of the ice sheet illustrate the steep margins simulated by CISM2 (Fig. 6). Most PMIP3 models produce ice sheets in CISM2 that are broadly similar in extent but lower than ICE-6G C and the various combinations of PDD factors produce largely similar ice sheets. MRI-CGCM3 has the largest spread across the combinations of PDD factors, indicating the feedback driving the northward collapse from the southern margin, as described above, is sensitive to
- the PDD factors. The steeper simulated southern margins result in total ice volumes that exceed that of 25 ICE-6G\_C. In most cases, the simulated bedrock depression is in good agreement with ICE-6G\_C along the transect. The agreement in bedrock depression among the simulations and ICE-6G\_C indicates the CISM2 is capturing the balance between mantel relaxation and the load of the ice sheets. Exceptions are

FGOALS-g2 and MPI-ESM-P that place the southern margin at too low a latitude which depresses bedrock further south.

After 5000 simulated years, seven CISM2-GCM pairs perform well by simulating positive mass balance that largely aligns with the ice-covered areas prescribed in the GCMs. There is particularly good agreement along the southern margins in the CCSM4, GENMOM, and GISS-E2-R CISM2 simulations (Fig. 7), which each produce mass balance equilibrium lines that correspond to the GCM ice-land boundary. In each of these simulations the ice sheet boundary extends slightly beyond that of the GCM, where the CISM2 simulated mass balance is strongly negative. July air temperatures in FGOALS-g2 and IPSL-CM5A-LR are 1 – 3 °C cooler than GENMOM or GISS-E2-R along southern margin of the CIS (not shown), in agreement with Pollard (2000) that the mass balance along the margin is very sensitive to small temperature differences. The GCMs that simulated below freezing July temperatures in Beringia (CCSM4, FGOALS-g2, GISS-E2-R, MIROC-ESM) display expanded areas of positive mass balance as ice volume develops. The two GCMs that fail to produce the correct extent of the LIS in CISM2 (CNRM-CM5 and MRI-CGCM3) have greatly reduced areas of positive mass balance relative to their GCM

prescribed ice sheet extent.

# 3.4 Volume and sea level equivalent

Our experimental design and use of fixed climatologies to drive CISM2 limits the degree to which the simulated total North American ice volumes are directly comparable to reconstructions, but it is useful to analyze total ice volume among models and PDD factors (Fig. 8). The CCSM4, GENMOM, GISS-E2-

- R, and IPSL-CM5A-LR driven simulations produce ice volumes that are in reasonable agreement (within  $\pm 10$  %) with ICE-6G\_C over a range of PDD factors. Every GENMOM PDD simulation has NAIS volumes within  $\pm 10\%$  ICE-6G\_C. All combinations of PDD factors yield ice volumes > 110% of ICE-6G\_C in the FGOALS-g2, MIROC-ESM and MPI-ESM-P based simulations. Simulations that produce NAIS changes in volume >90 m of sea level equivalent (SLE) are likely unrealistically high given that
- the CMIP5/PMIP3 estimated total global LGM SLE is 121.5 m, which includes contributions from Eurasia and Antarctica (Abe-Ouchi et al., 2015). Since most of the CISM2-GCM combinations simulate the extent of the LIS reasonably well, the excess ice is attributed to the spurious simulation of ice in

Beringia and steep idealized margins. No combination of PDD factors lead to ice volumes within  $\pm 10\%$  of ICE-6G\_C in the CNRM-CM5 or MRI-CGCM3 simulations. The suite of PDD factors produce an average range of 4.3 x  $10^6$  km<sup>3</sup> of ice volume from the seven GCMs that simulate a reasonably sized LIS with CISM2, indicating total ice volume is sensitive to the choice of PDD factors even though ice sheet

area largely is not.

The simulated CISM2 ice volumes did not reach steady state after 5000 years. The seven GCMs driving CISM2 that simulated the LIS area reasonably well continued to accumulate to unrealistically large volumes when the CISM2 was integrated for up to 50,000 model years. The ice volume values reported here are therefore sensitive to the time period of analysis. Our use of 5000 years was chosen as

the approximate time period when ice sheet area stabilizes in most simulations. Allowing the ice sheet volume to come to equilibrium is a function of CISM2 parameters such as the sliding scheme, basal traction coefficients, viscosity, and flow factor as opposed to local surface mass balance from the GCMs.

#### **4** Discussion

#### 4.1 GCM snow cover and albedo

The nine GCMs include different Land Surface Models (LSMs); although, CCSM4 and FGOALSg2 use different versions of the Community Land Model (see Table 9.A.1 in, Flato et al., 2013). The seasonal cycle of snow and albedo in the LGM simulations differ substantially among the LSMs. Regions with 'run away snow' occur over the ice sheet where summer melt is insufficient to offset snowfall so year-round snow continues to accumulate. The presence of deep snowpack alters the seasonality of 20 albedo, net radiation and the sensible and latent heat fluxes. To prevent the models from accumulating snow indefinitely, most GCMs impose a maximum limit on either snow depth or snow amount, which varies by model. For example, GENMOM limits snow depth to 10 m, whereas CCSM4 limits snow amount to 1000 kg/m<sup>2</sup>.

Most of the GCMs simulate year-round snow cover over substantial areas of the interior of the 25 LIS (Fig. 9). CNRM-CM5 is the exception due to its anomalously warm air temperature which results in lower fractional snow coverage in March, whereas all the other models maintain ~100% fractional

coverage. The GISS-E2-R March snow cover map indicates masking artifacts similar to CNRM-CM5, with snow depth set to zero in Hudson Bay, and in parts of Beringia (i.e. grid cells that are ocean in the piControl experiment). The regions with July snow cover correspond well to regions with positive mass balance in CISM2 (Fig. 7), including the presence of July snow cover in Beringia. Both CCSM4 and MRI-CGCM3 display low July snow cover along the southern margin of the ice sheet, corresponding to above freezing temperatures.

The methods used to determine albedo in LSMs ranges from specified values that are spatially and temporally invariant to computations that account for variations in solar declination angle, snow age, snow optical properties, ice thickness, and the presence of surface melt ponds (Pollard, 2000). It is beyond

- the scope of this paper to detail the albedo calculation in each GCM; however, some broad observations are insightful. The GCMs display a wide range of July albedo over NAIS (Fig. 10). The six GCMs that simulate the largest ice volumes in CISM2 also have high albedos corresponding to the ice-covered areas prescribed in the GCM. In the six GCMs, the July albedos are generally > 0.7 on the ice-covered grid cells. Half of these six models display spatially uniform albedos (IPSL-CM5A-LR, MIROC-ESM, MPI-
- ESM-P), whereas albedo is spatially varying in the other half (CCSM4, GENMOM, GISS-E2-R). CNRM-CM5 and MRI-CGCM3 display lower albedos than the other PMIP3 models, particularly in areas that are snow-free. The southern portion of the ice sheet in MRI-CGCM3, for example, has an albedo of ~0.12, a value twice that of seawater and half that of land cells. The low albedos on snow-free ice cells are possibly due to the LSMs approximating surface melt ponds, which are less reflective than either snow or ice. In
- contrast, GENMOM has a simple representation of surface liquid and melt ponds such that visual waveband albedos linearly decrease between -5 °C and 0 °C with minima of 0.6 for snow and 0.5 for ice (Pollard, 2000).

#### 4.2 Ice in Beringia

Of the nine PMIP3 models, four support extensive areas of ice whereas the remaining five are largely ice-free (Fig. 7). Adjacent pollen reconstructions for northern Alaska indicate a mixed response of warmer (cooler) and wetter (drier) LGM climate relative to present (Bartlein et al., 2011). Some PMIP2 models displayed warmer-than-present surface temperatures in Beringia due to altered circulation patterns