# Peer review of "Application of an ice sheet model to evaluate PMIP3 LGM climatologies over the North American ice sheets"

_Climate of the Past, 2017_

## Referee Comment (RC1) · Anonymous Referee #1 · 6 Sep 2017

Introduction:

The authors present an attempt to evaluate climate model results for the last glacial maximum (LGM) produced as part of the Paleoclimate Modelling Intercomparison Project 3 (PMIP3) using an ice sheet system model (ISM). The evaluation is largely based on results from the surface mass balance model of the ISM, which employs a positive-degree-day (PDD) melt model.

Main comments:

The biggest problem with the present manuscript and my reason to advice rejection is the ice dynamical aspect of the modelling. As the authors write themselves P10,6, the

relaxation time of 5000 years is not enough for the model to reach steady state. The chosen period is clearly too short for the ice flow to sufficiently respond to the imposed SMB forcing. This means that the ice sheet is in an arbitrary state after 5000 years as it relaxes from the assumed initial reconstructed geometry to balance with the imposed forcing. This is a fundamental flaw in the experimental setup.

Although the ice sheet model is described as thermodynamic, I see no evidence that the ice temperature is evolved or even initialised. This would have to be clarified. However, it is important to realise the main difference between the options discussed to arrive at an initial state (btw. I don't understand the second option for initialisation at P4,1 without a reference). While the result of the first approach is a fully self-consistent (thermo-)dynamic ice sheet model state, this is not the case for the given choice of imposing a reconstructed geometry.

Furthermore, differences in the SMB forcing between climate models imply that each GCM generates its individual response time scale dependent e.g. on the spatial SMB gradients. It is possible that the North American ice sheets were never really in balance with the climate during the LGM (something to discuss), but assuming an arbitrary period to evolve from an arbitrary initial state is certainly not an acceptable solution to this problem.

Another problem with the present setup (that would at least need to be acknowledged) is that the "coupling" between climate and ice sheet model is reduced to the lapse rate effect on temperature. The further away the ice sheet geometry evolves from the ice sheet that was prescribed for the GCM, the less reliable are the climatic fields entering the calculations. This is in particular a problem where the land type changes e.g. from ice sheet to land cover, or vice versa.

I believe a study relying entirely on the SMB component of the ISM would arrive nearly at the same conclusions as the present manuscript. The statements P4,16 are clearly ignoring any flow of the ice, which indicates that the dynamic aspect of the ice sheet

model is not really considered in the discussion anyway. Further support for that approach could be drawn from the July panels of Figure 2, which give a good impression where a feasible ice sheet can exist. In that case, however, the SMB model (here PDD) would have to be treated with much more detail and a clearer correspondence with the underlying GCM results would be in place. An interesting additional check would be what climates the GCMs produce as present-day conditions (positive SMB for Greenland, ...) to distinguish models that are generally warm biased from models that are warm biased only at the LGM but OK for the present and the same for cold biases for the two periods.

I am afraid this is unfortunate timing, but some, if not much of this work will likely be superseded by the CP discussion paper by Niu et al. https://doi.org/10.5194/cp-2017-105.

Minor comments:

Abstract: The long form of PMIP3 should be "Paleoclimate Modelling Intercomparison Project", not "Palaeoclimate Modelling Intercomparison"

P13,13 something wrong with this sentence

Discussion and reference to Ziemen et al. in the introductions seems appropriate. doi:10.5194/cp-10-1817-2014

---

## Referee Comment (RC2) · I. Rogozhina (Referee) · 20 Oct 2017

Reviewer: Irina Rogozhina, University of Bremen

The manuscript of Adler and Hostetler evaluates the performance of eight climate simulations of the Last Glacial Maximum included in the PMIP3 and one in-house climate simulation over North America and Greenland. They use a thermodynamic ice sheet model CISM to argue that two GCMs produce excessively warm conditions over the region that could not possibly sustain realistic geometries of the former North American ice sheet complexes. Although the other seven climatologies are generally consistent with existing reconstructions of the LGM ice sheet extents in North America, four of

them produce an undocumented ice cover across Beringia. To evaluate the robustness of their conclusions, the authors assess the sensitivity of their modeled ice sheet geometries to the choice of degree-day factors in the positive degree day model and an ice flow enhancement factor.

Regardless of its rather simple modeling approach, I find this study insightful. In particular, this work shows that modeled climate conditions are often inconsistent with the prescribed ice sheet boundary conditions and that ice sheet models could serve as valuable tools for evaluation of climate models in areas where paleoclimate proxy data are absent. Thus, I believe it merits publication in CP, albeit after moderate revisions. I have a number of suggestions on how to improve the analysis presented and fill the gaps in the model description.

The model description is incomplete and has to be improved. Even though the model setup is partly adopted from Gregiore et al. (2012), all parameter values and a short description of different model components should be included in the present manuscript (e.g., the description of the GIA model, underwater ice scheme and basal sliding law are missing; there is no mention of the geothermal flux forcing; daily temperature standard deviation value used in the PDD scheme is not provided, etc.). It is not clear how the authors initialize their ice sheets. While their initial ice configurations are based on the ICE-6G reconstruction, it is not clear what the authors prescribe inside of these ice sheets – how are the initial ice temperatures and ages derived? I suggest that in their revised manuscript the authors carefully detail their simulation design and methods.

Improvement of the main set of experiments:

The authors have suggested an interesting approach to an ice sheet model initialization in an attempt to start simulations from ice configurations, which are broadly consistent with the boundary conditions prescribed in climate models. We all struggle with the choice of model initialization strategies, when only equilibrium GCM outputs of certain time slices are available, whereas ice sheet configurations during these time intervals

represent products of the preceding long-term climate history and a gradual ice sheet buildup. At this instance I disagree with reviewer 1 that the glacial index approach is a consistent way to initialize an ice sheet model, since it implies that (i) climate conditions of let's say MIS3 can be derived as some intermediate state between the LGM and present-day climate fields; (ii) climate conditions at the peak MIS5e could be extrapolated beyond the warmth of the Holocene period using the LGM and present-day climate fields; (iii) Greenland (or Antarctic) ice core reconstructions can reproduce air temperature (and precipitation) variability across the entire globe. Climate proxy data and geological evidence from several regions around the world show a very different picture, for example Central Asia and the Russian Arctic (including Beringia discussed in the text). Glacial expansions in the Kara Sea region during the last glacial cycle were obviously asynchronous with the rest of the continental glaciations in the Northern Hemisphere (Patton et al., 2015). Hence, I feel it is important that the ice sheet modeling community (concerned with both former and present-day ice sheet reconstructions) starts looking for new methods to initialize their ice sheet models. The suggestion of Adler and Hostetler is a good initial step towards this goal. Although their idea is interesting, it may be meaningless from the physical point of view, especially if the authors do not ensure that the ice sheet model has had sufficient time to recover from the initial shock arising from unrealistic initial thermal and dynamical ice regimes. In essence, their initial ice sheet configurations are empty shells, which are inconsistent with the physical laws underpinning the ice sheet model used. Since some of the North American ice sheets were very thick at the LGM (comparable in thickness with the present-day Greenland and Antarctic ice sheets), they may need a rather long initialization time (several tens of thousands of years) to forget this initial non-physical shock. I believe that the integration time the authors have adopted is insufficient to achieve this goal. One way to demonstrate that the initial shock is not altering the general conclusions of the study is to perform an additional test forced by one of the GCM model outputs, which are not entirely unrealistic (e.g., GENMOM or MPI). In this test the ice sheet model should be initialized over a time period of 20-50 thousand years

using a time-invariant climate forcing and keeping the ice thickness fixed as is done for the present-day ice sheets (e.g., Pattyn, 2010). After this initialization, the authors could run a forward simulation with a free ice thickness evolution and compare their modeled extents to the ones presented in the current version of the manuscript. Another way would be to run their simulations for a longer time period (20 thousand years at the least). According to the current state of knowledge, a large portion of North America had been buried under ice sheets over a period of more than 50 thousand years during the last glacial cycle. Hence, longer integration times would not hurt.

Improvement of sensitivity tests:

The authors have performed an extensive analysis of the uncertainties in the degree-day factors but have disregarded potential effects of the daily temperature standard deviation (in the PDD scheme), which is rather uncertain in nature (e.g., Fausto et al., 2009; Seguinot, 2013; Wake and Marshall, 2014). Their choice of a snow-to-rain fraction (100% vs 0%) and a meltwater retention scheme could be also objected. Should not these be included in their sensitivity tests? I realize that such tests would add many simulations to the story but running them with a resolution of 40 km and a SIA-only model is more than feasible. All sensitivity tests could be placed in the supplement (including the tests of sensitivity to degree-day factors)

Typos and concerns: Page 1: PMIP3 -> correction suggested by reviewer 1 Page 3, line 15: An awkward statement: 40-km North American domain - refomulate Page 7, line 23: prescribe -> prescribed Page 8, line 10: Do you really mean "steep domes" here? Maybe "sleep margins"? Page 8, lines 11-12: This statement is difficult to believe. Could we see a proof of it in the supplement? Page 8, line 28: Typo: mantle Page 10, line 1: Why are margins idealized? Is it not a product of the initialization chosen? Page 13, line 14: delete one "is"

References:

Patton, H., K. Andreassen, L. R. Bjarnadóttir, J. A. Dowdeswell, M. C. M. Winsborrow, R. Noormets, L. Polyak, A. Auriac, and A. Hubbard (2015), Geophysical constraints on the dynamics and retreat of the Barents Sea ice sheet as a paleobenchmark for models of marine ice sheet deglaciation, Rev. Geophys., 53, 1051–1098, doi:10.1002/2015RG000495

Pattyn, F.: Antarctic subglacial conditions inferred from a hybrid ice sheet/ice stream model, Earth Planet. Sc. Lett., 295, 451–461, doi:10.1016/j.epsl.2010.04.025, 2010.

---

## Author Comment (AC1) · 8 Nov 2017

Response notation as follows:

Referee comment (RC1): standard text

Author comment (AC): *italicized text*

RC1: The biggest problem with the present manuscript and my reason to advice rejection is the ice dynamical aspect of the modelling. As the authors write themselves P10,6, the relaxation time of 5000 years is not enough for the model to reach steady state. The chosen period is clearly too short for the ice flow to sufficiently respond to

the imposed SMB forcing. This means that the ice sheet is in an arbitrary state after 5000 years as it relaxes from the assumed initial reconstructed geometry to balance with the imposed forcing. This is a fundamental flaw in the experimental setup.

*AC: Our experimental design was chosen so that we could compare the simulated areal footprint of the ice sheet to that of the driving GCM. The modeled surface mass balance and areal extent stabilize surprisingly quickly, with the exception of Beringia which can take tens of thousands of simulated years to stabilize (Figure R1). We do note in the text (P10,6) that the ice sheet is not in steady state thermodynamically, but we felt a short simulation period was sufficient for our purposes. We apply the ice sheet model as a sensitivity tool to validate the GCM climatology. To address the concerns from both reviewers, our revised manuscript will analyze the models after 50,000 years of simulation where the ice sheet is in or near steady state. This will most likely require the modification or removal of the discussion of ice volume (Section 3.4) as comparing simulated volume after 50,000 years of constant climate forcing relative to ICE-6G is not appropriate.*

RC1: Although the ice sheet model is described as thermodynamic, I see no evidence that the ice temperature is evolved or even initialised. This would have to be clarified.

*AC: Both reviewers are correct, in that we omitted a description of how the ice sheet temperature was initialized. Ice temperature was initialized at 0 °C as the ICE-6G reconstruction provides no information on ice temperature or flow vectors. This is a limitation of the experimental design. It is clear from further analysis (Figure R2) that it takes tens of thousands of years for the model to recover from the initial non-physical shock of initializing with 0 °C ice, to use RC2's wording. This will be addressed in the revised manuscript and Figure R2 will be included in SI. Even though the lower levels of the ice sheet stabilize after 50,000 years, temperature continues to drift in the upper levels in some models. We feel that the model temperature profile is sufficiently spun up after 50,000 years and that will be an appropriate window for our revised analysis.*

RC1: However, it is important to realise the main difference between the options discussed to arrive at an initial state (btw. I don't understand the second option for initialisation at P4,1 without a reference). While the result of the first approach is a fully self-consistent (thermo-)dynamic ice sheet model state, this is not the case for the given choice of imposing a reconstructed geometry.

*AC: A climate index (or glacial index) method of initialization is desirable from an ice sheet modeling perspective because it allows the ice sheet to develop slowly over a long period of time and produces a thermodynamically self-consistent model state. However, as RC2 correctly points out, simply linearly interpolating the climate forcing between LGM and pre-industrial is not realistic. Interpolating between LGM and pre-industrial climate states is a stopgap when very long transient glacial cycle simulations are not available, but GCM snapshots are (i.e. PMIP3). Initializing CISM2 with large ice sheets with no internal structure is also physically unrealistic, but the climate forcing is consistent with the initial ice sheet geometry. Both initialization methods have unphysical limitations, but we choose the later method because we are interested in if the PMIP3 model simulated climatology would support the large North American ice sheets they are driven by, as opposed to creating realistic ice sheet inception and chronologies (i.e. marine isotope stage events). We will clarify these points in the revised manuscript.*

RC1: Furthermore, differences in the SMB forcing between climate models imply that each GCM generates its individual response time scale dependent e.g. on the spatial SMB gradients. It is possible that the North American ice sheets were never really in balance with the climate during the LGM (something to discuss), but assuming an arbitrary period to evolve from an arbitrary initial state is certainly not an acceptable solution to this problem.

*AC: It is clear at this point that an analysis window of 5000 years was too short, but we disagree with the reviewer that the initial state is arbitrary. The initial geometry and volume corresponds to the ICE-6G reconstruction, which is an approximation for the*

*static ice sheets used to drive the GCMs. The benefit of this is the forcing temperature and precipitation climatology are consistent with the initial ice sheet geometry. As we noted above, initializing the ice sheet model with no internal structure is a limitation, but we feel this will be mitigated by analyzing the ice sheet model output near steady state.*

RC1: Another problem with the present setup (that would at least need to be acknowledged) is that the "coupling" between climate and ice sheet model is reduced to the lapse rate effect on temperature. The further away the ice sheet geometry evolves from the ice sheet that was prescribed for the GCM, the less reliable are the climatic fields entering the calculations. This is in particular a problem where the land type changes e.g. from ice sheet to land cover, or vice versa.

*AC: This is implicit with one-way offline coupling (P5,4-8), but we will clarify the point in our revised manuscript. This limitation applies to both a climate index method and our own reconstruction based initialization method and can only truly be addressed through two-way coupling where a dynamic ice sheet model is included in an Earth System Model or EMIC.*

RC1: I believe a study relying entirely on the SMB component of the ISM would arrive nearly at the same conclusions as the present manuscript.

*AC: In a previous version of our manuscript we did compare the SMB at the first step of the ice sheet model to the SMB after 5000 years (essentially what the reviewer suggests). The contours of the Laurentide and Cordilleran Ice Sheets can largely be arrived at from SMB alone, but this is not the case for CNRM-CM5 and MRI-CGCM3 and is not always the case for ice development in Beringia.*

RC1: The statements P4,16 are clearly ignoring any flow of the ice, which indicates that the dynamic aspect of the ice sheet model is not really considered in the discussion anyway.

*AC: These statements are overly simplistic, and will be revised in our manuscript, but we do not feel these statements are indicative that the overall paper ignores ice dynamics. See P7,12-16; P8,15-19; P8,20-P9,2 for examples.*

RC1: Further support for that approach could be drawn from the July panels of Figure 2, which give a good impression where a feasible ice sheet can exist. In that case, however, the SMB model (here PDD) would have to be treated with much more detail and a clearer correspondence with the underlying GCM results would be in place.

*AC: As with initial SMB, July temperature corresponds to locations of simulated ice presence in broad terms, but not in the details or in models that have marginal climatologies.*

RC1: An interesting additional check would be what climates the GCMs produce as present-day conditions (positive SMB for Greenland, ...) to distinguish models that are generally warm biased from models that are warm biased only at the LGM but OK for the present and the same for cold biases for the two periods.

*AC: Evaluating the piControl derived ice simulation is outside the scope of our paper which uses CISM2 as sensitivity tool to validate the PMIP3 LGM climatology.*

RC1: I am afraid this is unfortunate timing, but some, if not much of this work will likely be superseded by the CP discussion paper by Niu et al. https://doi.org/10.5194/cp-2017-105.

*AC: The Niu et al (2017) paper largely focuses on the COSMOS model with limited discussion of PMIP3 GCM forcing. Their paper developed pseudo glacial volume histories via a glacial index, which was not our goal. We do not feel our analysis and discussion of PMIP3 LGM simulations (particularly our discussion on albedo) are in any way superseded by the Niu et al (2017) paper.*

Please also note the supplement to this comment:

https://www.clim-past-discuss.net/cp-2017-102/cp-2017-102-AC1-supplement.pdf

---

## Author Comment (AC2) · 8 Nov 2017

Response notation as follows:

Referee comment (RC2): standard text

Author comment (AC): *italicized text*

RC2: The manuscript of Adler and Hostetler evaluates the performance of eight climate simulations of the Last Glacial Maximum included in the PMIP3 and one in-house climate simulation over North America and Greenland. They use a thermodynamic ice sheet model CISM to argue that two GCMs produce excessively warm conditions over

the region that could not possibly sustain realistic geometries of the former North American ice sheet complexes. Although the other seven climatologies are generally consistent with existing reconstructions of the LGM ice sheet extents in North America, four of them produce an undocumented ice cover across Beringia. To evaluate the robustness of their conclusions, the authors assess the sensitivity of their modeled ice sheet geometries to the choice of degree-day factors in the positive degree day model and an ice flow enhancement factor.

Regardless of its rather simple modeling approach, I find this study insightful. In particular, this work shows that modeled climate conditions are often inconsistent with the prescribed ice sheet boundary conditions and that ice sheet models could serve as valuable tools for evaluation of climate models in areas where paleoclimate proxy data are absent. Thus, I believe it merits publication in CP, albeit after moderate revisions. I have a number of suggestions on how to improve the analysis presented and fill the gaps in the model description.

*AC: We thank you for your constructive feedback and for noting that we are using CISM2 as a tool for evaluating sensitivity rather than trying to reconstruct glacial histories. A number of your suggestions prompted us to do additional simulations (which will be included in the revised paper SI), which largely indicate that the results in our manuscript are robust.*

RC2: The model description is incomplete and has to be improved. Even though the model setup is partly adopted from Gregiore et al. (2012), all parameter values and a short description of different model components should be included in the present manuscript (e.g., the description of the GIA model, underwater ice scheme and basal sliding law are missing; there is no mention of the geothermal flux forcing; daily temperature standard deviation value used in the PDD scheme is not provided, etc.). It is not clear how the authors initialize their ice sheets. While their initial ice configurations are based on the ICE-6G reconstruction, it is not clear what the authors prescribe inside of these ice sheets – how are the initial ice temperatures and ages derived? I suggest

that in their revised manuscript the authors carefully detail their simulation design and methods.

*AC: In hindsight, our description of the ice sheet model components and description of how we initialize the reconstructed ice sheet geometry require more detail. We had used the default value of 5 °C for the daily temperature standard deviation (dd_sigma in CISM2). We provide more information on this parameter below. As indicated in our response to RC1, the ice temperature was initialized at 0 °C everywhere as the ICE-6G reconstruction provides no information on ice temperature or flow vectors. This is a limitation of the experimental design. It is clear from further analysis (Figure R2) that it takes tens of thousands of years for the model to recover from the initial non-physical shock of initializing with 0 °C ice, so our use of 5000 years was too short to capture thermodynamic stability. In our revised manuscript, we will analyze the CISM2 simulations after 50,000 years.*

RC2: The authors have suggested an interesting approach to an ice sheet model initialization in an attempt to start simulations from ice configurations, which are broadly consistent with the boundary conditions prescribed in climate models. We all struggle with the choice of model initialization strategies, when only equilibrium GCM outputs of certain time slices are available, whereas ice sheet configurations during these time intervals represent products of the preceding long-term climate history and a gradual ice sheet buildup. At this instance I disagree with reviewer 1 that the glacial index approach is a consistent way to initialize an ice sheet model, since it implies that (i) climate conditions of let's say MIS3 can be derived as some intermediate state between the LGM and present-day climate fields; (ii) climate conditions at the peak MIS5e could be extrapolated beyond the warmth of the Holocene period using the LGM and present-day climate fields; (iii) Greenland (or Antarctic) ice core reconstructions can reproduce air temperature (and precipitation) variability across the entire globe. Climate proxy data and geological evidence from several regions around the world show a very different picture, for example Central Asia and the Russian Arctic (including Beringia discussed

in the text). Glacial expansions in the Kara Sea region during the last glacial cycle were obviously asynchronous with the rest of the continental glaciations in the Northern Hemisphere (Patton et al., 2015). Hence, I feel it is important that the ice sheet modeling community (concerned with both former and present-day ice sheet reconstructions) starts looking for new methods to initialize their ice sheet models. The suggestion of Adler and Hostetler is a good initial step towards this goal. Although their idea is interesting, it may be meaningless from the physical point of view, especially if the authors do not ensure that the ice sheet model has had sufficient time to recover from the initial shock arising from unrealistic initial thermal and dynamical ice regimes. In essence, their initial ice sheet configurations are empty shells, which are inconsistent with the physical laws underpinning the ice sheet model used. Since some of the North American ice sheets were very thick at the LGM (comparable in thickness with the present-day Greenland and Antarctic ice sheets), they may need a rather long initialization time (several tens of thousands of years) to forget this initial non-physical shock. I believe that the integration time the authors have adopted is insufficient to achieve this goal. One way to demonstrate that the initial shock is not altering the general conclusions of the study is to perform an additional test forced by one of the GCM model outputs, which are not entirely unrealistic (e.g., GENMOM or MPI). In this test the ice sheet model should be initialized over a time period of 20-50 thousand years using a time-invariant climate forcing and keeping the ice thickness fixed as is done for the present-day ice sheets (e.g., Pattyn, 2010). After this initialization, the authors could run a forward simulation with a free ice thickness evolution and compare their modeled extents to the ones presented in the current version of the manuscript. Another way would be to run their simulations for a longer time period (20 thousand years at the least). According to the current state of knowledge, a large portion of North America had been buried under ice sheets over a period of more than 50 thousand years during the last glacial cycle. Hence, longer integration times would not hurt.

*AC: We agree with your assessment that linearly interpolating between LGM and pre-industrial is not a realistic way to treat the climate forcing. Our goal is to determine if the*

*climate forcing from the PMIP3 LGM models would support the ice sheets they included in their boundary conditions. We did not manipulate the temperature and precipitation fields and we initialized the ice sheet geometry close to that used in the GCMs so that the climate fields and ice sheet were initially consistent. We acknowledge that this meant initializing the ice sheets with no internal structure, which is a limitation of our approach. We will clarify these points in the revised manuscript.*

*It does not appear that CISM2 can be integrated with a fixed ice sheet height when using the SIA and PDD schemes. This option is only available when using the high-order ice dynamic core, which is outside the scope of our study. We feel most of the concerns from RC1 and RC2 can be addressed by evaluating our model results after 50,000 years of simulation, when the ice sheet is at or near steady state (see Figures R1 and R2).*

RC2: Improvement of sensitivity tests: The authors have performed an extensive analysis of the uncertainties in the degree-day factors but have disregarded potential effects of the daily temperature standard deviation (in the PDD scheme), which is rather uncertain in nature (e.g., Fausto et al., 2009; Seguinot, 2013; Wake and Marshall, 2014). Their choice of a snow-to-rain fraction (100% vs 0%) and a meltwater retention scheme could be also objected. Should not these be included in their sensitivity tests? I realize that such tests would add many simulations to the story but running them with a resolution of 40 km and a SIA-only model is more than feasible. All sensitivity tests could be placed in the supplement (including the tests of sensitivity to degree-day factors)

*AC: We had not previously explored the sensitivity of our results to the uncertainties with the daily temperature standard deviation parameter (dd_sigma). We analyzed daily temperature time series from GENMOM and found that the daily standard deviation varies spatially and seasonally. Like observations from modern Greenland, GENMOM does show more daily temperature variability in the winter than summer, but the simulated LGM values in North America are much higher than those from Greenland observations. The annually averaged values range from 7 - 8 °C. The CISM2 annual*

*PDD scheme does not allow sigma to vary in space or by season (without changing the model code, which is beyond the scope of our work). To test this parameter, we drove CISM2 with 50 years of daily GENMOM output, using the daily PDD scheme, and compared the results with results obtained from a range of uniform sigma values. (Figure R3). This test indicates that an annual PDD sigma value of 6 °C matches the ice sheet volume and area simulated with the daily PDD reasonably well. We will include Figure R3 and a discussion of daily temperature variability in the main text of our revision. We have also updated all our CISM2-PMIP3 simulations to use a dd_sigma value of 6 °C; although, the areal extent is largely the same as our previous simulations (dd_sigma value of 5 °C), with the exception of Beringia in a few models (Figure R4).*

*The revised manuscript will include Figure R5 in the SI, which demonstrates the sensitivity of CISM2 to the daily temperature standard deviation (dd_sigma), the fraction of snow that refreezes (wmax) and the ice flow factor. We will use these extra sensitivity experiments to support statements such as P8, 11-12. We will keep the 12 PDD factor sensitivity tests in the main paper, as it provides context to Figure 4, which we feel is important because it demonstrates that no combination of PDD factors result in a reasonable LIS when forced by CNRM-CM5 or MRI-CGCM3 and that ice sheet extent is largely insensitive to the choice of PDD factors. By adopting 50,000 year simulations, the discussion of ice sheet volume (Section 3.4) will likely need to be removed, which means the PDD sensitivity results will be limited to Figures 4 and 6. In the CISM2 implementation of annual PDD, the snow-to-rain fraction is not a tunable parameter. With the addition of a discussion on daily temperature standard deviation (Figure R3) and the low sensitivity of melt water refreeze (shown in Figure R5), we will have discussed and addressed the range of PDD parameters.*

Please also note the supplement to this comment:
https://www.clim-past-discuss.net/cp-2017-102/cp-2017-102-AC2-supplement.pdf

**Supplement:**

[Figure]

Figure R1. CISM2 time evolution of North American ice area and volume. Sea level equivalent is calculated as the eustatic change from converting land ice to water, using 910 and 1028 kg m$^{-3}$ densities for ice and water respectively, assuming an ocean area of 360 768 576 km$^2$. This figure will be included in the revised manuscript SI.

[Figure]

Figure R2. Time evolution of the temperature profile over 100,000 simulated years. Temperature is spatially averaged over a 20x20 grid cell box in the center of the Laurentide Ice Sheet [lower left 102.28°W, 56.68°N; lower right 89.93°W, 56.77°N; upper right 88.89°W, 63.64°N; upper left 103.91°W, 63.51°N]. Sigma level = 1 is the bottom of the ice sheet and sigma level = 0 is the top. This figure will be included in the revised manuscript SI.

[Figure]

Figure R3. CISM2 sensitivity tests comparing 50 years of daily input from GENMOM using the daily PDD scheme to monthly climatology input from GENMOM using the annual PDD scheme with multiple parameterizations of daily temperature standard deviation ($\sigma$). $PDD_{ice}$ = 16 mm d$^{-1}$ °C, $PDD_{snow}$ = 4 mm d$^{-1}$ °C, 60% of snowmelt refreezes. Additional daily PDD parameters (e.g. snow density and fraction of rainfall to snowfall) are set to CISM2 defaults. All other parameters follow Table 1 in the main text. Ice thickness is sampled after 50,000 years of simulation. This figure will be included in the revised manuscript main text.

[Figure]

Figure R4. Sensitivity tests of ice sheet thickness using different parameterizations of daily temperature standard deviation (dd_sigma) after 50,000 years of simulation. The annual PDD scheme is used with $PDD_{ice}$ = 16 mm $d^{-1}$ °C, $PDD_{snow}$ = 4 mm $d^{-1}$ °C. Parameters follow Table 1 in the main text.

[Figure]

Figure R5. Sensitivity tests of ice sheet thickness using different parameterizations of a) daily temperature standard deviation (dd_sigma), b) the fraction of snow that refreezes (wmax) and the c) ice flow factor. GENMOM is used as input to all simulations and ice thickness is sampled after 50,000 years of simulation. The annual PDD scheme is used with $PDD_{ice}$ = 16 mm d$^{-1}$ °C, $PDD_{snow}$ = 4 mm d$^{-1}$ °C. Unless otherwise specified, parameters follow Table 1 in the main text. This figure will be included in the revised manuscript SI.